# Behavioral problems in anxious youth: Cross-sectional and prospective associations with reinforcement sensitivity and parental rejection

Leonie J. Kreuze[1]*, Peter J. de Jong[1], Nienke C. Jonker[1], Catharina A. Hartman[2], Maaike H. Nauta[1]

**1** Department of Clinical Psychology and Experimental Psychopathology, University of Groningen, Groningen, The Netherlands, **2** Interdisciplinary Center Psychopathology and Emotion Regulation, University of Groningen, Groningen, University Medical Center Groningen, The Netherlands

* l.j.kreuze@rug.nl

**Data Availability Statement:** Data Availability: Under the General Data Protection Regulation (GDPR), our dataset is considered pseudonymized

## Abstract

A subsample of children and young people (CYP) with anxiety disorders presents with comorbid behavioral problems. These CYP have greater impairment in daily life, profit less from current treatments, and have an increased risk for continued mental problems. We investigated two potential explanations for these comorbid behavioral problems. First, high punishment sensitivity (PS) may lead to a strong inclination to experience threat, which may not only elicit anxiety but also defensive behavioral problems. Second, behavioral problems may arise from high reward sensitivity (RS), when rewards are not obtained. Behavioral problems may subsequently elicit parental rejection, thereby fueling anxiety. We used a cross-sectional (age = 16.1, $N = 61$) and prospective (age = 22.2, $N = 91$) approach to test the relationship between PS/RS and comorbid behavioral problems. Participants were a subsample of highly anxious CYP from a large prospective cohort study. PS/RS were indexed by a spatial orientation task. We also investigated the prospective association between behavioral problems and anxiety at 6-year follow-up, and the proposed mediation by parental rejection. PS and RS showed no cross-sectional or prospective relationships with comorbid behavioral problems in highly anxious CYP. Yet, behavioral problems in adolescence showed a small prospective relationship with anxiety in young adulthood, but this was not mediated nor moderated by parental rejection. No evidence was found for PS/RS being involved in comorbid behavioral problems in anxious CYP. Findings point to comorbid behavioral problems as potential factor contributing to the further increase of anxiety.

## Introduction

Anxiety disorders are among the most prevalent disorders in children and young people (CYP) [1]. In apparent conflict with the stereotypical expression of anxiety disorders, a subgroup of anxiety disordered CYP also meets criteria for a disruptive behavioral disorder [2–4].

rather than anonymized, and is still regarded as personal data. When participants were invited to the cohort more than 20 years ago, they were not asked to give informed consent to make their personal data publicly available in pseudonymized form. As a result of this, legal and ethical restrictions prevent the authors from making data from the TRAILS Study publicly available. Data and syntasx are available upon request from the TRAILS data manager (trails@umcg.nl). Detailed information about the participation agreements with TRAILS participants is available from the ethics committee; Central Committee on Research Involving Human subjects (CCMO; tc@ccmo.nl). For more information about accessing data from the TRAILS Study, please see https://www.trails.nl/en/hoofdmenu/data/data-use.

**Funding:** This research is part of the TRacking Adolescents'Individual Lives Survey (TRAILS). Participating centers of TRAILS include various departments of the University Medical Center and University of Groningen, the Erasmus University Medical Center Rotterdam, the University of Utrecht, the Radboud Medical Center Nijmegen, and the Parnassia Bavo group, all in the Netherlands. TRAILS has been financially supported by various grants from the Netherlands Organization for Scientific Research (NWO), ZonMW, GB MaGW, the Dutch Ministry of Justice, the European Science Foundation, BBMRI-NL, the participating universities, and Accare Center for Child and Adolescent Psychiatry. The first author was supported by a PhD fund from the faculty of Behavioural and Social sciences from the University of Groningen. TRAILS has been financially supported by grants from the Netherlands Organization for Scientific Research NWO (Medical Research Council program grant GB-MW 940-38-011; ZonMW Brainpower grant 100-001-004; ZonMw Risk Behaviour and Dependence grants 60-60600-97-118; ZonMw Culture and Health grant 261-98-710; Social Sciences Council medium-sized investment grants GB-MaGW 480-01-006 and GB-MaGW 480-07-001; Social Sciences Council project grants GB-MaGW 452-04-314 and GB-MaGW 452-06-004; NWO large-sized investment grant 175.010.2003.005; NWO Longitudinal Survey and Panel Funding 481-08-013 and 481-11-001; NWO Vici 016.130.002 and 453-16-007/2735; NWO Gravitation 024.001.003), the Dutch Ministry of Justice (WODC), the European Science Foundation (EuroSTRESS project FP-006), the European Research Council (ERC-2017-STG-757364 en ERC-CoG-2015-681466), Biobanking and Biomolecular Resources Research Infrastructure BBMRI-NL (CP 32), the Gratama foundation, the

CYP with anxiety and comorbid behavioral problems were found to benefit less from current treatment approaches [5,6], to have greater impairment in daily life [7–10] and to have an increased risk for continued mental health problems in adulthood [6,11]. Insight in the risk factors of this comorbidity may provide clues to improve their situation.

One factor that might contribute to comorbid behavioral problems is punishment sensitivity. There is ample evidence that CYP with heightened punishment sensitivity have an increased risk of developing anxiety symptoms [12–15]. People with high punishment sensitivity have a relatively strong inclination to interpret ambiguous situations in a threatening way [16–18]. This heightened punishment sensitivity might also make them more vulnerable for developing comorbid behavioral problems. That is, perceived threat may not only elicit anxiety, but also defensive anger/oppositional behaviors. In line with this, a recent study in non-clinical adolescents found that punishment sensitivity was associated with increased anger responses to scenario's reflecting common anger eliciting situations, and that this association was mediated by threat interpretations [19]. This defensive response is especially expected in situations with high levels of immediate perceived threat where people feel that they cannot avoid this threat [20,21]. Because people with high punishment sensitivity will be more inclined to experience threat, they may also be more inclined to respond with reactive aggression/anger in ambiguous situations that they interpret as threatening [22]. Therefore, punishment sensitivity might be a risk factor for developing comorbid behavioral problems in anxiety disordered CYP.

A second factor that might heighten the risk for developing comorbid behavioral problems in anxiety disordered CYP is high reward sensitivity. Multiple studies have indicated an association between reward sensitivity and behavioral problems. More specifically, reward sensitivity has been associated with, trait anger in non-clinical students [23,24], self-reported conduct problems in clinical adolescents [25], self-reported verbal and physical aggression in non-clinical students [23], and self-reported hostility in non-clinical students [24]. People with high reward sensitivity are highly motivated to gain rewards, more responsive to reward, and have more attention to rewarding cues in the environment. Therefore, people with high reward sensitivity have higher reward expectancies in ambiguous situations that may involve potential rewards [16,26]. This may result in the person taking action to gain the reward [27], and may have beneficial effects in daily life when these rewards are indeed obtained [28]. However, given their high reward expectancy, they are also more prone to detect non-reward (rewards with a lower than expected frequency or lower level of reward), which may lead to anger and oppositional behaviors out of frustrative non-reward [22,24,26,29,30]. In line with this, it was found that reward sensitivity was associated with increased anger responses to scenario's reflecting common anger eliciting situation, and that this association was mediated by non-reward interpretations in a non-clinical sample of adolescents [19]. Thus, high reward sensitivity might make CYP with anxiety disorders vulnerable for developing comorbid behavioral problems.

In the current study, we examined if indeed high punishment and reward sensitivity were associated with the strength of comorbid behavioral problems in highly anxious CYP. In order to identify a certain characteristic as a risk factor, this characteristic should precede the outcome of interest [31]. Longitudinal designs are therefore most suitable for studying potential risk factors. Therefore, in the current study we not only took a cross-sectional approach, but also looked at prospective associations while using a behavioural procedure (Spatial Orientation Task, [32]) to assess punishment and reward sensitivity. In the entire focus group ($N_{cross-sectional}$ = 696, $N_{longitudinal}$ = 598)) of the TRAILS cohort, we failed to find meaningful cross-sectional and prospective associations between punishment sensitivity (as indexed with the SOT) and anxiety symptoms and between reward sensitivity (as indexed with the SOT) and behavioral problems in CYP [33]. The non-significant association between reward

Jan Dekker foundation, the participating universities, and Accare Centre for Child and Adolescent Psychiatry.

**Competing interests:** The authors have declared that no competing interests exist.

sensitivity and behavioral problems in this entire focus group may be explained by previous research that found that for non-clinical CYP, reward sensitivity may be actually associated with positive social and environmental functioning [28]. The association between reward sensitivity with negative emotions and behavioral problems may especially hold for a clinical group. (Also, the predicted association between punishment sensitivity and behavioral problems was not tested in this previous study. This pattern of punishment sensitivity leading to anxiety and behavioral problems may be especially relevant for highly anxious CYP. Especially these highly anxious CYP may react with anger/oppositional behaviors to escape from threatening situations, whereas this is not the case for low anxious children. Therefore, our current predictions specifically apply to a highly anxious sample, as was used in the current study.

Comorbid behavioral problems in CYP may also indirectly contribute further to the maintenance of CYP's anxiety. Oppositional behaviors in CYP have been associated with parental rejection [34–36]. In turn, rejective parenting is associated with anxiety in CYP [37]. In this way, anxiety disordered CYP with comorbid behavioral problems may get stuck in a loop when their parents show rejective responses. The impact of parental rejection on anxiety in CYP may be especially pronounced for CYP that are more sensitive for punishment/rejection. A second aim of this study was therefore to investigate whether behavioral problems in adolescence increase the risk for having higher anxiety symptoms in young adulthood, whether this association was mediated by parental rejection, and whether this was especially the case for individuals with high punishment sensitivity.

In short, in the current study we first investigated cross-sectionally whether indeed behavioral problems in highly anxious adolescents were associated with (i) heightened punishment sensitivity and/or (ii) heightened reward sensitivity. Using a longitudinal approach, we secondly examined whether this heightened punishment sensitivity and/or reward sensitivity was prospectively associated with behavioral problems in highly anxious adults six years later. Finally, we investigated whether having behavioral problems in adolescence was associated with having higher anxiety symptoms in young adulthood, whether this association was (partly) mediated by parental rejection, and whether this was especially the case in individuals with relatively high punishment sensitivity (as measured in their adolescence).

## Method

This study was preregistered on Open Science Framework, the preregistration can be found via https://osf.io/4q7br/.

### Participants

The current study is embedded in the Tracking Adolescent's Individuals Lives Survey (TRAILS). TRAILS is a large prospective population study of Dutch adolescents coming from the five northern municipalities in the Netherlands including both rural and urban areas. Children born between 1 October 1989 and 30 September 1990 from two northern municipalities and children born between 1 October 1990 and 30 September 1991 from the remaining three northern municipalities form the TRAILS cohort. At baseline (T1), 2230 children were included, with assessments taking place in 2001 and 2002 [38,39]. Written informed consent was obtained from all adolescents and their parents.

The current study reports on data from the third (T3), fourth (T4) and fifth (T5) assessment waves [39,40]. Data collection during T3 took place between 2005 and 2007, 1816 adolescents participated (81% of the initial sample at T1) with a mean age of 16.3. Data collection during T4 ran from October 2008 to September 2010, 1881 adolescents participated (84% of the initial sample) with a mean age of 19.1 years. The fifth wave (T5) was conducted in 2012 and 2013;

1778 adolescents participated (80% of the initial sample). Participants were then between 21 and 24 years of age with a mean age of 22.3 years. In the current study we used a behavioral procedure to measure punishment and reward sensitivity (SOT, [32]), and questionnaires to assess anxiety symptoms, behavioral problems, and parental rejection.

The SOT was part of the assessment at T3, and was the first task in a series of laboratory tasks that were performed in addition to the general assessments during T3. For these time-intensive laboratory tasks a focus group of 744 participants was contacted, 715 (96%) of these agreed to participate. This focus group is overrepresented by adolescents with a high risk of mental health problems. High risk was based on temperament (high frustration and fearful-ness, low effortful control), lifetime parental psychopathology (depression, anxiety, addiction, antisocial behavior or psychoses) and/or living in a single parent family. Of this focus group, 66% had at least one of these risk factors. The focus group was complemented with a random selection from the low-risk TRAILS participants (see also [41]). It is possible to represent the TRAILS distribution in this focus cohort by means of sampling weights [38].

In the current study, we focused on clinically anxious CYP. Given that the TRAILS study was conducted in a large cohort of adolescents with low sampling bias, we expected that in accordance with previous findings in the literature on prevalence rates of anxiety [1,42] the 90[th] percentile scoring adolescents on the anxiety symptoms measure of the Revised Child Anxiety and Depression Scale-Child version (RCADS-C; [43,44]; Dutch version: [45] at T3 represents a high anxious sample of clinical relevance. Similarly, we selected a high anxious young adult sample (T5) by selecting the 90[th] percentile scoring young adults on the anxiety subscale of the Adult Self Report (ASR; [46]). A cut-off score of .85 (mean item score) for the 90[th] percentile on the RCADS anxiety scale was calculated based on the entire TRAILS sample at mean age 16.3 (n = 1657) and a cut-off score of .75 (mean item score) for the 90[th] percentile on the ASR anxiety subscale was calculated based on the entire TRAILS sample at mean age 22.3 (n = 1499). For the cross-sectional part of the study, we selected participants who completed the SOT at T3 and who were highly anxious based on the RCADS cut-off (*n* = 61). For the prospective part of the study, we selected participants who completed the SOT at T3 and who were highly anxious participants based on the ASR cut-off score at T5 (*n* = 91).

The moderated mediation analysis was restricted to participants who completed the SOT (T3), the questionnaires measuring anxiety at T3 and T5, the questionnaire measuring behavioral problems at T3, and the questionnaire measuring parental rejection at T4 (*n* = 560). We checked whether only including participants that filled out all the required measures for the moderated mediation analysis led to a biased sample, by comparing the included participants with the missing participants on important variables at T3 (anxiety and behavioral problems) and T4 (parental rejection). No significant differences were found. Therefore, we expect that our sample was not biased and given that power is high enough we continued the analysis with this selected sample. Characteristics of the samples can be found in Table 1

## Measures

**Anxiety.** At T3, anxiety was assessed using the Revised Child Anxiety and Depression Scale-Child version (RCADS-C). The RCADS-C [44] (Dutch version: [45] consists of 47 items that are rated on a 4-point scale from 0 = never to 3 = always. The RCADS measures symptoms of DSM-IV anxiety disorders and depression in children from the ages of 7 to 19. The RCADS-C has six subscales; separation anxiety disorder, social phobia, generalized anxiety disorder, obsessive compulsive disorder, panic disorder, and major depressive disorder. In the current study, we calculated a mean item score based on items of only those subscales that correspond to the primary anxiety disorders of children as classified in the DSM-5; separation anxiety disorder (7

**Table 1. Characteristics of the samples.**

| | T3 | T5 |
|---|---|---|
| Highly anxious cross-sectional sample (N = 61) Associations PS and RS with behavioral problems M(SD) range/percentage | | |
| Age | 16.1 (0.5) 14.9–17.5 | - |
| Gender % female | 80 | - |
| *Anxiety* RCADS YSR | 1.1 (0.2) 0.9–2.0 0.8 (0.4) 0.3–1.8 | - |
| *Behavioral problems* YSR | 0.5 (0.3) 0.1–1.2 | - |
| Highly anxious prospective sample (N = 91) Associations PS and RS with behavioral problems M(SD) range/percentage | | |
| Age | - | 22.2 (0.7) 21.2–24.1 |
| Gender % female | - | 67 |
| *Anxiety* RCADS ASR | - - | - 1.1 (0.3) 0.9–1.7 |
| *Behavioral problems* ASR | - | 0.5 (0.3) 0.0–1.5 |

Sample moderated mediation (N = 560) Association of behavioural problems with anxiety, mediated by parental rejection, moderated by PS

| | M(SD) range/percentage | M(SD) range/percentage M(SD) range/percentage |
|---|---|---|
| | **T3** | **T4** **T5** |
| Age | 16.1 (0.6) 14.7–18.1 | 19.0 (0.5) 18.0–20.4 22.2 (0.6) 21.0–24.1 |
| Gender % female | 54 | 54 |
| *Anxiety* | 0.4 (0.3) 0.0–1.7 | 0.4 (0.4) 0.0–1.7 |
| RCADS | 0.3 (0.3) 0.0–1.8 | . |
| YSR | 0.3 (0.2) 0.0–1.2 | 0.2 (0.2) 0.0–1.5 |
| ASR | | 1.5 (.4) 1.0–4.0 |
| *Behavioral problems* | | |
| YSR | | |
| ASR | | |
| Embu rejection | | |

*Note*. RCADS = Revised Child Anxiety and Depression Scale-Child version, YSR = Youth Self Report, ASR = Adult Self Report.

PS = attentional proneness components of punishment sensitivity.

RS = attentional proneness components of reward sensitivity.

items, Cronbach's $\alpha$ = .63), social phobia (9 items, Cronbach's $\alpha$ = .86), generalized anxiety disorder (6 items, Cronbach's $\alpha$ = .79), and panic disorder (9 items, Cronbach's $\alpha$ = .77).

At T3, anxiety was also assessed with the anxiety subscale of the Youth Self Report (YSR). The YSR consists of 112 items (scored on a 3-point scale from 0 = not true to 2 = very/often true) on behavioral and emotional problems in the past 6 months ([46]. The mean item score of the DSM-IV anxiety subscale of the YSR (6 items, Cronbach's $\alpha$ = 0.65) was included only in the moderated mediation analysis of this study to control for anxiety symptoms at T3. For this analysis the YSR was preferred over the RCADS, because of the YSR's similarity with the outcome measure (the Adults Self Report) used in the moderated mediation analysis. At T5, anxiety was assessed using the adult version of the Adult Self Report (ASR) which consist of 102 items (scored on a 3-point scale from 0 = not true to 2 = very/often true) on behavioral and emotion problems in the past 6 months [46]. The mean item score of the DSM-IV anxiety subscale of the ASR (7 items, Cronbach's $\alpha$ = .76) was included in the current study.

**Behavioral problems.** At T3, behavioral problems were assessed with the Youth Self Report (YSR) [46], using the aggressive behavior subscale (17 items scored on a 3 point scale from 0 = not true to 2 = very/often true; Cronbach's $\alpha$ = .81), the mean item score on this subscale was used in the current study. At T5 behavioral problems were assessed with the adult version of the Youth Self Report, namely the Adult Self Report (ASR), using the mean item score on the aggressive behavior subscale (15 items, measured on a 3-point scale from 0 = not true to 2 = very/often true; Cronbach's $\alpha$ = .84).

**Perceived parental rejection.** At T4, perceived parental rejection was assessed with 4 items stemming from the parental rejection scale of the Egna Minnen Beträffande Uppfostran (a Swedisch acronym for my Memories of Upbringing) (EMBUC, [47]) assessing rejective behavior from the mother (4 items measured on a 4-point scale from 1 = no, never to 4 = yes, most of the time, Cronbach's $\alpha$ = .67) and rejective behavior from the father (4 items Cronbach's $\alpha$ = .70). A combined parental rejection score was calculated by calculating a mean rejection score out of the mother and father rejection scores, the mean item score was used in this study. In case a child reported on only one parent, the score related to that parent was included.

**Sensitivity to punishment and reward (Spatial Orientation Task (SOT)).** The method of the SOT task is the same as described in (31) given that the current study focuses on a subsample of the entire focus group of this previous study. The SOT is a motivated game that was developed to examine individuals' inclination to direct and hold their attention to cues signaling reward and punishment [32]. In line with previous research in the context of eating disorder and substance misuse [48–51], we used the inclination to direct and hold attention to cues signaling reward as measures of reward sensitivity and the inclination to direct and hold attention to cues signaling punishment and non-punishment as measure of punishment sensitivity.

Participants have to respond as quickly as possible to a neutral target that is preceded by a cue in order to gain points or to avoid losing points. They have to press the 'b' key as soon as they see the target. Their score is displayed in the middle of the screen. There are two types of games, in losing games, participants lose 10 points if they respond too slowly, and their score remains unchanged if they respond sufficiently fast, whereas in winning games, participants win 10 points if they respond sufficiently fast, and their score remains unchanged if they respond too slowly. At the beginning of the task, participants were told that those with the highest scores in the winning games would win an attractive prize (i.e., a balloon ride) and that an extremely low score on the losing games would result in having to redo the task until their performance was good enough. Participants lose 10 points regardless of the game type when they respond when no target appeared (catch trials) or before the target has appeared. The task consists of four losing and four winning games, which are alternated every two games. Each game consists of 32 cued, 16 uncued and 8 catch trials that are presented randomly. Before these eight games, participants get four practice games (two losing and two winning) each consisting of 6 cued, 6 uncued and 2 catch trials. The task was performed on an Intel Pentium 4 CPU computer with a Philips Brilliance 190 P monitor and run by E-prime software version 1.1. (Psychology Software Tools Inc., Pittsburgh, Pennsylvania). Participants were seated 50 cm away from the screen and responses were collected on the computers' keyboard (39). S1 Fig in the supplement shows the critical elements of the SOT.

## Components of the SOT task

Cued or uncued: Each trial starts with the appearance of two vertical black bars on a white background, left and right of the participant's score that is presented in the middle of the screen. This score was set to zero at the beginning of each block. A new trial is signaled by the current score disappearing from the screen for 200 ms after which it reappeared. After a 250 ms delay, a cue replaced one of the two black bars. Then after a delay of either 250 (short delay) or 500 ms

(long delay) the target appeared either centered within the cue or centered within the remaining black bar on the other side of the screen. Consistent with previous studies using the SOT [41,49–51], the presentation of the target with either 250 or 500 ms delay provides the opportunity to examine the relative importance of early (short delay) attentional processes and attentional processes that allow for some regulatory control (long delay). When the target appears in the cue, the trial is called a cued trial, when the target appears in the uncued black bar, the trial is called an uncued trial. This cue operates as a signal of reward/non-punishment or punishment/non-reward by indicating the change of winning or losing points.

**Signals of reward/non-punishment and punishment/non-reward.** The task included two different cues that could precede the target; a blue arrow pointing upwards and a red arrow pointing downwards. Participants were informed that both cues indicated the probable location of the target, with 2/3 of the targets appearing in the cued location. It was explained that in general the blue cue was a signal for having a high change of responding fast enough (fast enough 75% of the time when cued, 25% of the time when uncued), whereas the red cue was a signal for having a high change of a too slow response (fast enough 25% of the time when cued, 75% of the time fast enough when uncued). So in general the blue arrow becomes a signal of reward (in winning games) or non-punishment (in losing games) because the chance of being fast enough is high, and the red arrow becomes a signal of non-reward (in winning games) or punishment (in losing games) because the chance of not being fast enough is high. Lastly, participants were informed that occasionally would be trials where no target appeared.

**Feedback.** After 500 ms in each response (or 1 second in the case of catch trial), the cue and target are removed, and the two black bars appear again. A feedback signal is given below the score. Both the blue upward arrow and the red downward arrow were also used as a feedback signal. The blue arrow signaled a fast-enough response on targeted trials or a correct non-response on catch trials. The red arrow signaled a too slow response on targeted trials or an inappropriate response on catch trials. After 250 ms the score is updated if necessary.

**Calculation of cutoffs for fast and slow responses.** At the end of each game, the participant's median reaction time and standard deviation based on all trials in that game were calculated to compute cutoffs for fast and slow responses in the following game of the same type. For the first two practice blocks a fixed cutoff of 350 ms was used since no personalized cutoffs were available for these blocks. During easy trials (cued blue or uncued red) responses were labeled sufficiently fast when they were faster than participant's median reaction time plus 0.55 times the standard deviation. During hard trials (uncued blue or cued red) responses were labeled sufficiently fast when they were faster than participants' median reaction time minus 0.55 times the standard deviation. Further, since reaction times tend to be about 25 ms slower after a short cue delay time than after a long cue delay time, 12 ms were added to the median reaction time for short-delay trials and 12 ms were subtracted from the median reaction time for long-delay trials (30)). This was done after the median reaction time for that game was calculated.

## Calculation of proneness to attend to (non)reward/(non)punishment

In line with [33] and [52], the proneness to attend to rewarding/punishing cues was indexed by the cue validity effect for cues signaling reward/punishment. The difference in reaction time between the cued and uncued location represents the cue validity effect. This cue validity effect reflects the inclination to direct attention to cues predicting punishment or reward. The mean reaction time to cued blue trials (signaling high chance of reward in winning games/non-punishment in losing games) was subtracted from the mean reaction time to uncued blue trials, where in general people are expected to be slower on uncued trials, leading to a positive difference score. To the extent that participants are more prone to attend to rewarding/non-

punishing cues, this difference will be larger, since they will be relatively slow when the target appears in the uncued condition compared to the cued condition, when the cues signal a high chance of reward/non-punishment. Similarly, the cue validity effect for cues signaling punishment/non-reward were computed by subtracting the mean reaction time to cued red trials (signaling a high chance of punishment in losing games/non-reward in winning games) from the mean reaction time to uncued red trials, where in general people are expected to be slower on uncued trials, leading to a positive difference score. To the extent that participants are more prone to attend to punishing/non-rewarding cues, this difference will be larger, since they will be relatively slow when the target appears in the uncued compared to the cued condition, when the cues signal a high chance of punishment/non-reward.

In order to take individual differences in reaction times into account when calculating the cue validity effects, we subtracted the individual's mean reaction time on the practice trials on either cued or uncued trials from the corresponding mean scores. This subtraction reduces the correlation between the components (reaction time of cued trials and reaction time of uncued trials) of the cue validity effects and thereby improves the reliability of attentional proneness measures [33]. See Table 2 for the calculations of the cue validity effects and the reliability estimates. The reliability estimates of the controlled cue validity effects indicate that each of the calculated cue-validity effects that we included in our analyses (in bold) had acceptable reliability.

## Indices of punishment and reward sensitivity used in the current study

The SOT assesses the attentional proneness to cues signaling punishment and reward and is thought to reflect behavioral output of individuals' punishment and reward system [32,53]. The attentional system provides the mechanism for detecting and monitoring the

**Table 2. Calculation of the cue validity effects controlled for mean reaction time, the interpretation of the cue validity effects, and reliability estimates in the cross-sectional and prospective sample.**

| Reward and Punishment indices | Calculation | Interpretation | Cue delay time* | Reliability estimate Spearman-Brown coefficient controlled for individual's mean reaction time in cross-sectional (c) and prospective sample (p) |
|---|---|---|---|---|
| **Winning game**<br>Cue validity effect for cues signaling reward | (mean RT uncued blue trials–mean RT uncued practice trials)–(mean RT cued blue trials–mean RT cued practice trials) | High score: stronger cue validity effect for cues signaling reward | 250 ms | **.79 (c) .80 (p)** |
| | | | 500 ms | **.76 (c) .80 (p)** |
| Cue validity effect for cues signaling non-reward | (mean RT uncued red trials–mean RT uncued practice trials)–(mean RT cued red trials–mean RT cued practice trials) | High score: stronger cue validity effect for cues signaling non-reward | 250 ms | .77 (c) .74 (p) |
| | | | 500 ms | .70 (c) .76 (p) |
| **Losing game**<br>Cue validity effect for cues signaling punishment | (mean RT uncued red trials–mean RT uncued practice trials)–(mean RT cued red trials–mean RT cued practice trials) | High score: stronger cue validity effect for cues signaling punishment | 250 ms | **.78 (c) .83 (p)** |
| | | | 500 ms | .46 (c) .72 (p) |
| Cue validity effect for cues signaling non-punishment | (mean RT uncued blue trials–mean RT uncued practice trials)–(mean RT cued blue trials–mean RT cued practice trials) | High score: stronger cue validity effect for cues signaling non-punishment. | 250 ms | **.78 (c) .79 (p)** |
| | | | 500 ms | **.72 (c) .80 (p)** |

*Note.* The cue validity effects with a bold reliability estimate are included in the analyses. *The presentation of the target with either 250 or 500 ms delay provides the opportunity to examine the relative importance of early (250 ms) attentional processes and attentional processes that allow for some regulatory control (500 ms).

RT = reaction time.

environment for stimuli that are relevant to the motivational state of the organism [54]. People who are heightened punishment sensitive are motivated to avoid punishment and are therefore expected to have a stronger cue validity effect for cues signaling punishment, which is expected to be present with short cue delay but not with long cue delay [53,55–57] Additionally, previous research also indicated that highly anxious individuals showed an enhanced cue validity effect for cues signaling non-punishment, especially with long cue delay, which may reflect a tendency to seek safety [32].

People who are heightened reward sensitive are motivated to obtain rewards and are therefore expected to have a stronger cue validity effect for cues signaling rewards, which might already be present with short cue delay and may be more pronounced with long cue delay since then also more voluntary processes can play a role [16,18,58]. In short, this led to the following measures: (i) punishment sensitivity is indexed by a stronger cue validity effect for cues signalling punishment with short cue delay and a stronger cue validity effect for cues signaling non-punishment with both short and long cue delay, (ii) reward sensitivity is indexed by a stronger cue validity effect for cues signalling reward with both short and long cue delay.

## Procedure

This study reports on data of a large prospective cohort study; a cross -sectional as well as prospective approach were taken. The Dutch (national) Central Committee on Research Involving Human Subjects (CCMO) approved the study. Anxiety and behavioral problems were measured with self-reports during the regular assessments at T3 and T5, perceived parental rejection was assessed with self-report during the regular assessment at T4, which took place at the TRAILS offices. The laboratory tests at T3 including the SOT were carried out at selected locations in or near the place of residence of participants, in a room with blinded windows that was sound attenuated. Test-assistants received extensive training in order to optimize standardization of the experimental session. See Fig 1 for a timeline of the study.

## Analytic plan

To test if behavioral problems in highly anxious CYP were associated with (i) heightened punishment sensitivity and/or (ii) heightened reward sensitivity, we first calculated bivariate correlations between the cue validity effects and behavioral problems. In case significant correlations as predicted in our hypotheses would be found between behavioral problems and the cue validity effects, multiple regression analyses were planned in step 2, to see whether their relationships with behavioral problems are additive or not. The cue validity effects from the losing games would be included in a regression model to look at the associations between proneness for punishing and non-punishing cues with behavioral problems and the cue validity effects from the winning games would be included in a regression model to look at associations between proneness for rewarding and non-rewarding cues with behavioral problems. If we would find that both the cue validity effects of the losing and winning games predicted behavioral problems, we would perform a regression analysis on that outcome variable including all eight cue validity effects (from both the winning and losing games) to see whether they explain the same variance or have (also) unique contributions.

To complement the results of the statistical analyses from step 1 and 2 following the common frequentist approach, we also reported results following the Bayesian approach. Hereby we aimed to increase the confidence in our results, and in case of non-significant findings, it provides us information about the strength of the evidence for the null-hypothesis. The Bayesian analyses were conducted with JASP (JASP Team, 2018). We conducted Bayesian correlational analyses using a default prior of 1 using a distribution that is uniform from -1 to 1 [59].

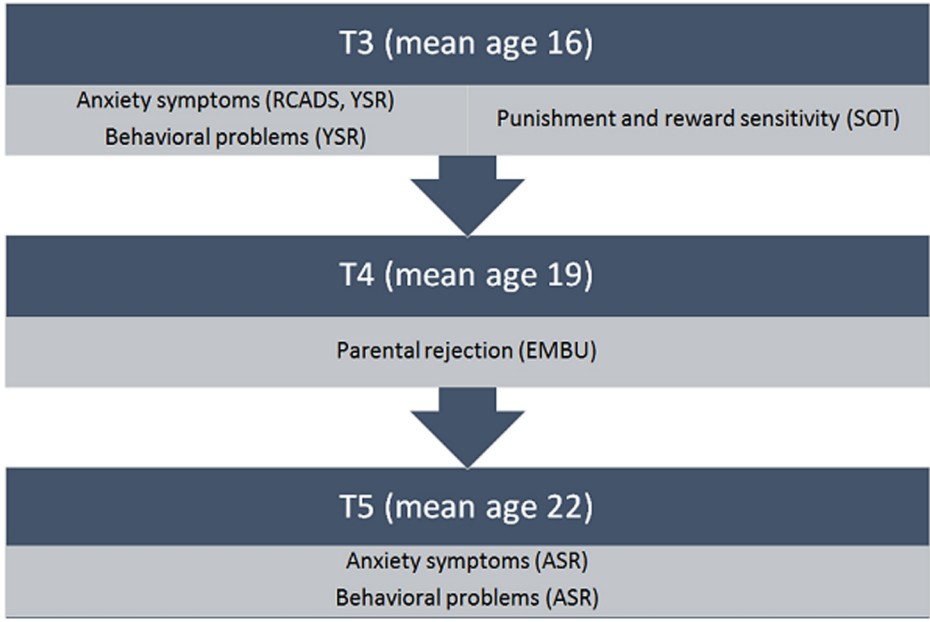

**Fig 1. Timeline and measures.** *Note.* Revised Child Anxiety and Depression Scale-Child (RCADS), Youth Self Report (YSR), Spatial Orientation Task (SOT), Egna Minnen Beträffande Uppfostran (EMBU), Adult Self Report (ASR).

To facilitate interpretation of the outcomes, BF01, which quantifies the evidence for the null hypotheses over the alternative hypothesis (there is no positive association between the cue validity effect and behavioral problems), was reported for our insignificant findings. BF10, which quantifies the evidence of the alternative hypothesis over the null hypothesis (there is a positive association between the cue validity effect and behavioral problems) was reported for our significant findings. A Bayes factor of 1 is considered no evidence, between 1 and 3 anecdotal/weak, between 3 and 10 moderate, between 10 and 30 strong, between 30 and 100 very strong, and more than 100 extremely strong evidence [60].

To test whether having behavioral problems in adolescence was associated with having higher anxiety symptoms in young adulthood, whether this association was (partly) mediated by parental rejection, and whether this was especially the case in individuals with relatively high punishment sensitivity (as measured in their adolescence), a moderated mediation analysis was conducted in SPSS using process models. We used Hayes model 14 were our dependent (y) variable was the ASR T5 anxiety score, our independent variable (x) was behavioral problems at T3, YSR T3 Anxiety was included as covariate to statistically control for anxiety level at T3, our mediator (m) was EMBU parental rejection (T4) and our moderator (v) was the cue validity effect for punishing cues with short cue delay. Anxiety in adolescence was included as covariate, since we wanted to investigate whether behavioral problems in adolescence are an additional risk for having anxiety symptoms in young adulthood, over and above anxiety in adolescence.

## Results

### Data reduction

In line with [41], trials of the SOT during which participants did not respond to the target were deleted, which resulted in deletion of 3.3% of the trials. Also, reaction times below 125 ms, which are expected to be anticipation errors, were deleted, resulting in the deletion of 8.5%

of the remaining trials. Furthermore, trials on which participants responded before the target appeared were removed, resulting in the deletion of 8.3% of the trials. Following, we selected the participants that fulfilled the inclusion criteria mentioned in the method section. The mean reaction times for each game type (winning and losing) and trial type (easy cue/hard cue and cued/uncued) were calculated after these deletions and selecting the participants and are presented in Tables 3, 4 and 5.

*Step 1*: Bivariate correlations were calculated between the cue validity effects, anxiety symptoms, and behavioral problems, see Table 6. The correlations of relevance for testing our hypotheses are reported in bold.

We would conduct separate regression analyses including cue-validity effects from either the winning or losing games when multiple significant cue validity effects would be found. However, given that only one cue-validity effect from the losing games correlated significantly with behavioral problems and one from the winning games, no regression analyses were conducted. Per hypothesis, the results of the correlational analyses are reported below.

**Behavioral problems in highly anxious adolescents are associated with heightened punishment sensitivity.** No significant correlation was found between behavioral problems in anxious adolescents and a stronger cue validity effect for cues signaling punishment with short cue delay. The Bayes factor indicated anecdotal/weak evidence in favor of H0 ($r = .15$, $p = .118$, $BF_{01} = 1.79$). A sensitivity analysis using a more conservative and lenient prior indicated that conclusions based on these analyses remained the same. Additionally, no significant correlations were found between behavioral problems in anxious adolescents and a stronger cue validity effect for cues signaling non-punishment with both short ($r = .12$, $p = .187$, $BF_{01} = 2.63$) and long cue delay ($r = .15$, $p = .129$, $BF_{01} = 1.93$). The Bayes factors showed anecdotal/weak evidence in favor of H0.

**Behavioral problems in highly anxious adolescents are associated with heightened reward sensitivity.** No significant correlation was found between behavioral problems in adolescence and a stronger cue validity effect for cues signaling reward with short cue delay ($r = .00$, $p = .505$, $BF01 = 6.32$), the Bayes factor indicates moderate evidence in favor of H0. However, a significant correlation was found for the cue validity effect for cues signaling reward with long cue delay ($r = .25$, $p = .027^*$, $BF10 = 1.90$), the Bayes factor showed there is only anecdotal/weak evidence in favor of a positive association.

**Punishment sensitivity is prospectively associated with behavioral problems in highly anxious adults six years later.** A significant correlation was found between behavioral problems in anxious young adults and a stronger cue validity effect for cues signaling punishment with short cue delay, however the Bayes factor showed there is only anecdotal/weak evidence for a positive association ($r = .19$, $p = .039^*$, $BF_{10} = 1.17$). No significant correlations were found between behavioral problems in young adulthood and a stronger cue validity effect for

**Table 3. Mean reaction times and standard deviations of the spatial orientation task in the cross-sectional sample (n = 61).**

| | Cued | | Uncued | |
|---|---|---|---|---|
| | Blue | Red | Blue | red |
| **Losing game** | | | | |
| Short cue delay time (250 ms) | 340 (46) | 366 (51) | 453 (87) | 465 (96) |
| Long cue delay time (500 ms) | 344 (54) | 381 (75) | 390 (78) | 381 (87) |
| **Winning game** | | | | |
| Short cue delay time (250 ms) | 338 (39) | 375 (45) | 463 (86) | 475 (87) |
| Long cue delay time (500 ms) | 357 (70) | 386 (70) | 397 (84) | 383 (68) |

**Table 4. Mean reaction times and standard deviations of the spatial oriental task in the prospective sample (n = 91).**

| | Cued | | Uncued | |
|---|---|---|---|---|
| | **Blue** | **Red** | **Blue** | **Red** |
| **Losing game** | | | | |
| Short cue delay time (250 ms) | 341 (46) | 368 (56) | 469 (77) | 489 (100) |
| Long cue delay time (500 ms) | 344 (62) | 373 (67) | 399 (84) | 395 (85) |
| **Winning game** | | | | |
| Short cue delay time (250 ms) | 343 (47) | 372 (51) | 481 (97) | 487 (95) |
| Long cue delay time (500 ms) | 350 (60) | 386 (67) | 404 (83) | 393 (79) |

cues signaling non-punishment with both short (*r* = .15, *p* = .084, BF01 = 1.64) and long cue delay (*r* = .09, *p* = .208, BF01 = 3.49). The Bayes factors indicated anecdotal/weak to moderate evidence in favor of H0.

**Reward sensitivity is prospectively associated with behavioral problems in highly anxious adults six years later.** No significant correlations were found between behavioral problems in young adulthood and a stronger cue validity effect for cues signaling reward with both short (*r* = .13, *p* = .108, BF01 = 2.02) and long cue delay (*r* = .15, *p* = .076, BF01 = 1.52). The Bayes factors indicated anecdotal/weak evidence in favor of H0.

Behavioral problems in adolescence are associated with having higher anxiety symptoms in young adulthood; this association is (partly) mediated by parental rejection, which is especially the case in individuals with relatively high punishment sensitivity.

*Simple mediation.* First, a simple mediation analysis was conducted to test whether the relationship between behavioral problems in adolescence (predictor) and anxiety in young adulthood (outcome variable) can be accounted for by parental rejection. In Table 2 the results of the mediation analysis are presented. Regressing behavioral problems in adolescence on anxiety in young adulthood indicated a significant total effect (path *c*), which remained significant in the full model (direct effect; path *c'*). Additionally, behavioral problems in adolescence significantly predicted parental rejection (path *a*;). However, parental rejection was not a significant predictor of anxiety in young adulthood (path *b*). The indirect effect of behavioral problems in adolescence on anxiety in young adulthood (path *ab*) showed a confidence interval including zero, indicating that parental rejection did not mediate the relationship between behavioral problems in adolescence and anxiety in young adulthood (see Table 7).

## Moderated mediation

As a next step, we investigated whether punishment sensitivity moderated the effect of parental rejection on anxiety in young adulthood and therefore this more complex model would help in explaining the association between behavioral problems in adolescence and

**Table 5. Mean reaction times and standard deviations of the spatial oriental task on cue validity effect for short cue delay in the losing game in the moderated mediation sample.**

| | Cued | | Uncued | |
|---|---|---|---|---|
| | **Blue** | **Red** | **Blue** | **Red** |
| **Losing game** | | | | |
| Short cue delay time (250 ms) | 327 (42) | 356 (49) | 455 (86) | 458 (93) |

*Note. n* = 560.

**Table 6. Bivariate correlations of cue validity effects with internalizing and behavioral problems at T3 and T5.**

| | 1 | 2 | 3 | 4 | 5 | 6 | 7 | 8 | 9 | 10 | 11 | 12 | 13 |
|---|---|---|---|---|---|---|---|---|---|---|---|---|---|
| 1 Anxiety T3 (RCADS) | | | | | | | | | | | | | |
| 2 Anxiety T3 (YSR) | .34* | | | | | | | | | | | | |
| 3 Behavioral problems T3 | .09 | -.07 | | | | | | | | | | | |
| 4 Anxiety T5 | .08 | .03 | .10 | | | | | | | | | | |
| 5 Behavioral problems T5 | .13 | .07 | .36* | .55* | | | | | | | | | |
| 6 parental rejection T4 | .09 | .08 | .05 | -.11 | .08 | | | | | | | | |
| 7 CV-reward short | .04 | .06 | **-.00** | .23* | **.13** | -.11 | | | | | | | |
| 8 CV-reward long | -.07 | .04 | **.25*** | .17 | **.15** | -.01 | .59* | | | | | | |
| 9 CV-nonreward short | -.06 | .09 | .15 | .17 | .15 | -.08 | .79* | .70* | | | | | |
| 10 CV-nonreward long | -.08 | .02 | .16 | .14 | .11 | -.06 | .54* | .66* | .66* | | | | |
| 11 CV-punishment short | -.05 | .12 | **.15** | .26* | **.19*** | -.14 | .78* | .67* | .85* | .56* | | | |
| 12 CV-punishment long | -.06 | .06 | **.31*** | .12 | .13 | -.10 | .55* | .73* | .67* | .71* | .64* | | |
| 13 CV-nonpunishment short | -.04 | -.02 | **.12** | .23* | **.15** | -.21* | .82* | .51* | .76* | .56* | .80* | .56* | |
| 14 CV-nonpunishment long | -.11 | .01 | **.15** | .15 | **.09** | -.14 | .74* | .77* | .72* | .65* | .76* | .75* | .69* |

*Note*. Correlations between T3 variables and Cue Validity (CV) effect variables are based on a sample size of $n = 61$, correlations with only T5 variables are based on a sample size of $n = 91$, correlations between variables of different measurement occasions are based on sample sizes varying from 83–88.

* p < .05. Correlations printed in bold are the tested correlations in this study.

anxiety in young adulthood. Table 3 depicts the results from this moderated mediation analysis. As can be seen in Table 3, the interaction between parental rejection and punishment sensitivity did not significantly predict anxiety in young adulthood (path $b_3$), indicating that punishment sensitivity did not moderate the association between parental rejection and anxiety in young adulthood. This is in line with the finding that the bootstrap confidence interval of the index of moderated mediation included zero. Thus, we did not find support that parental rejection mediated the association between behavioral problems in adolescence and anxiety in young adulthood, also not when taking punishment sensitivity into account (see Table 8 and Fig 2).

**Table 7. Mediation analysis for the association between behavioral problems in adolescence and anxiety in young adulthood via parental rejection.**

| | Path/effect | B | SE | t | p | 90% CI |
|---|---|---|---|---|---|---|
| Simple Regression Models[1] | | | | | | |
| $R^2 = .240$ $F (2,557) = 87.88$ p < .001 | c (total effect of BP on Anx) | .135 | .059 | 2.28 | .023* | 0.04; 0.23 |
| $R^2 = .066$ $F (2,557) = 19.54$ p < .001 | a (BP on PR) | .368 | .072 | 5.08 | < .001* | 0.25; 0.49 |
| Multiple Regression Model* | | | | | | |
| $R^2 = .240$ $F (3,556) = 58.67$ p < .001 | c' (direct effect of BP on Anx) | .127 | .061 | 2.09 | .037* | 0.03; 0.23 |
| | b (PR on Anx) | .023 | .035 | .65 | .516 | -0.04; 0.08 |
| | | *Effect* | *Boot SE* | | | *Boot CI* |
| | ab (indirect effect of BP on Anx through PR) | .008 | .015 | | | -0.01; 0.03 |

BP = Behavioral problems in adolescence.

Anx = anxiety in young adulthood.

PR = parental rejection.

Boot = bootstrap.

[1] model includes anxiety in adolescence as covariate.

* significant with α of .05.

As a typical harmful environment for adolescents, parental rejection could also be considered as a moderator in the relation between behavioral problems in adolescents and later anxiety symptoms in young adulthood. We therefore conducted a simple moderation analysis using Hayes model 1 with behavioral problems in adolescence as independent variable, anxiety in young adulthood as dependent variable and parental rejection as moderator. Anxiety in adolescence was included as covariate.

As can be seen in Table 9, no significant moderation effect was found. Therefore, we did not compute a more complex model including punishment- or reward sensitivity.

## Discussion

### Main findings

Our study investigated punishment and reward sensitivity as risk factors for developing comorbid behavioral problems in highly anxious CYP. We used a behavioral procedure that measures the proneness for punishing cues and non-punishing cues to index punishment sensitivity and the attentional proneness for rewarding cues to index reward sensitivity. We found a significant positive association between attentional proneness for punishing cues with short cue delay and behavioral problems in young adulthood, however the Bayesian analyses showed that the strength of the evidence for this finding should be considered inconclusive. None of the other predicted associations between punishment sensitivity indices and behavioral problems in anxious adolescents and anxious young adults were found. For reward sensitivity, we found a significant cross-sectional association between the proneness for rewarding cues with long cue delay and behavioral problems in anxious adolescents, however, the Bayesian analyses showed that the strength of the evidence for this finding should be considered inconclusive. None of the other predicted associations between reward sensitivity indices and behavioral problems in anxious adolescents and young adults were found. Furthermore, we found that behavioral problems in adolescence predicted anxiety symptoms in young adulthood, over and above the predictive value of anxiety symptoms in adolescence. We did not find evidence

**Table 8. Moderated mediation analysis for the association between behavioral problems in adolescence and anxiety in young adulthood via parental rejection, moderated by punishment sensitivity.**

| | Path/effect | B | SE | t | p | 90% CI |
|---|---|---|---|---|---|---|
| See Table 2 for pathways/effects c (total effect of BP on Anx) and a (BP on PR) | | | | | | |
| Multiple Regression Model[1] | | | | | | |
| $R^2$ = .046 F (1,558) = 26.86 p < .001 | c'(direct effect of BP on Anx) | .126 | .061 | 2.08 | .038* | 0.03; .23 |
| | $b_1$ (PR on Anx) | .042 | .040 | 1.05 | .292 | -0.02; 0.11 |
| | $b_2$ (PS on Anx) | .000 | .000 | 1.04 | .300 | 0.00; 0.00 |
| | $b_3$ (PR *PS on Anx) | .000 | .000 | -.91 | .364 | -0.00; 0.00 |
| | | *Effect* | *Boot SE* | | | *Boot CI* |
| | Index of moderated mediation | .000 | .000 | | | 0.00; 0.00 |

BP = Behavioral problems in adolescence.

Anx = anxiety in young adulthood.

PR = parental rejection.

PS = proneness for punishing cues with short cue delay.

Boot = bootstrap.

[1]model includes anxiety in adolescence as covariate.

*significant with α of .05.

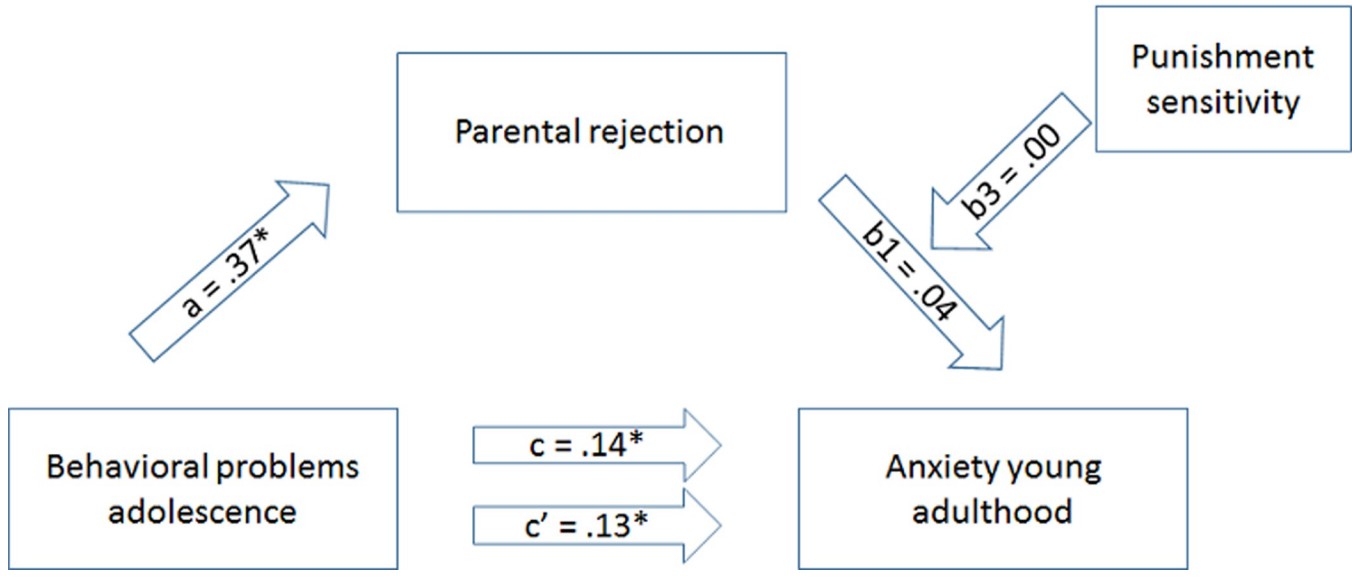

**Fig 2. Moderated mediation model.** Post-hoc analysis: moderation analysis.

for our expected mediation via parental rejection in this age-span. Also no moderation effect of parental rejection was found.

We expected that punishment and reward sensitivity would be risk factors for developing behavioral problems in highly anxious CYP. Given that we looked in highly anxious samples selected from a large representative cohort sample, we should have been able to find the predicted associations if they would exist. However only two of the ten expected associations between the behavioral measure of punishment and reward sensitivity with behavioral problems in anxious CYP were found. The evidence in favor of these positive associations was found to be weak, and therefore these findings are inconclusive. Also, false positive findings due to multiple testing cannot be ruled out. Overall, our study provided no evidence to support the hypothesis that punishment and reward sensitivity as measured by the SOT are risk factors for developing comorbid behavioral problems in anxious CYP. These findings extend previous

**Table 9. Moderation analysis for the association between behavioral problems in adolescence and anxiety in young adulthood, moderated by parental rejection.**

| | Anxiety in young adulthood | *B* | *SE* | *t* | *P* | *95% CI* |
|---|---|---|---|---|---|---|
| Multiple Regression Model[1] | | | | | | |
| $R^2 = .240\ F\ (2,555) = 44.45\ p < .001^*$ | *Constant* | .069 | .085 | 0.81 | .420* | -0.10; 0.24 |
| | *Behavioral problems* | .382 | .211 | 1.81 | .071 | -0.03; 0.80 |
| | *Parental rejection* | .083 | .059 | 1.40 | .16 | -0.03; 0.20 |
| | *Moderation effect behavioral problems* parental rejection* | -.169 | .134 | -1.26 | .207 | -0.43; 0.09 |
| | *Anxiety in adolescence* | *.550* | *.047* | *11.74* | *<0.001* * | *0.46; 0.64* |

BP = Behavioral problems in adolescence.

Anx = anxiety in young adulthood.

PR = parental rejection.

Boot = bootstrap.

[1]model includes anxiety in adolescence as covariate.

*significant with α of .05.

findings from a study in a sample of unselected CYP that also failed to find meaningful associations between punishment sensitivity and reward sensitivity (as indexed with the SOT) with anxiety symptoms and behavioral problems respectively (31).

## Attentional proneness to (non)-punishing cues and comorbid behavioral problems

People with high punishment sensitivity are expected to be more inclined to experience threat, and were, therefore, also expected to be more inclined to respond with reactive aggression/ anger in threat situations where they cannot avoid (18–20). We thus expected punishment sensitivity to be a risk factor for developing comorbid behavioral problems in anxiety disordered CYP. In line with this hypothesis, we did find a significant positive association between attentional proneness for punishing cues with short cue delay and behavioral problems in young adulthood. This might indicate that an inclination to experience threat in adolescence increases the chance of showing behavioral problems (which are expected to result as defensive response to threat) in young adulthood. However, the Bayesian analyses showed that the strength of the evidence for this finding should be considered inconclusive. Additionally, we would expect this association between an attentional proneness for punishing cues with short cue delay and behavioral problems to also arise cross-sectionally, however, this was not the case. Furthermore, we neither found the expected association between an attentional proneness to non-punishing cues and behavioral problems in adolescence nor in young adulthood. Therefore, the findings provided no clear evidence in support of our hypothesis. It might be that no association between punishment sensitivity and behavioral problems was found because the anxious CYP in our study might have been able to often avoid their anxiety-provoking situations. We expected that heightened punishment sensitivity would lead to behavioral problems due to defensive responses in threat situation where avoidance is hindered. However, if avoidance is possible, there is no need for defensive behavior and subsequently no association with behavioral problems is expected to arise. It might also be that other components of reward-and punishment sensitivity are more relevant for predicting comorbid behavioral problems in highly anxious CYP then the attentional component. Punishment and reward sensitivity are proposed to consist of different components that represent different aspects of these sensitivities [16,18,58,61], namely *responsivity* (how much punishment/reward influences your affect), the *motivation* to avoid punishment/approach reward, and the *attentional proneness* to punishing/rewarding cues. Since the current measure was restricted to the attentional component it cannot be ruled out that other components are still relevant for explaining comorbid behavioral problems in anxious persons. Self-report measures are well-suited to test the responsivity and motivation components of punishment and reward sensitivity. Multiple studies have found associations between self-report measures of punishment sensitivity with anxiety symptoms/disorders in CYP and adults [13,14] and between self-report measures of reward sensitivity and behavioral problems in CYP and adults [13,23–25]. Most of these previous studies were cross-sectional and did not focus on comorbid behavioral problems in highly anxious CYP. We do therefore not yet know whether these associations also exist prospectively and whether self-reported punishment and reward sensitivity are related to comorbid behavioral problems in anxious CYP. Additionally, the current study did not take into account that people with high reward sensitivity may not only display anger out of frustrative non-reward, but may also display more aggressive or disruptive behavior to get the reward when being hindered. Future research that differentiates between these two types of situations is necessary to disentangle these two different explanations. One option would be to employ a scenario approach with scenarios that refer to non-reward situations as well as situations in

which the reward could still be gained, and assess participants' emotions and behaviors in these situations as a function of their reward-sensitivity [e.g., 19].

## Attentional proneness to cues signaling reward and comorbid behavioral problems

We also expected that people with high reward sensitivity would be more prone to detect potential rewards in the environment, and would therefore also be more prone to detect non-reward, which might lead to comorbid behavioral problems out of frustrative non-reward [22,24,26,29,30]. In line with our expectation, we did find a significant cross-sectional association between the proneness for rewarding cues with long cue delay and behavioral problems in anxious adolescents. However, the Bayesian analyses showed that the strength of the evidence for this finding should be considered inconclusive. Additionally, we would expect this association to also be present prospectively, however, this was not the case. Also, we neither found the expected association between an attentional proneness for rewarding cues with short cue delay and behavioral problems in adolescence, nor in young adulthood. Therefore, the findings provided no clear evidence that attentional proneness to reward cues was a risk factor for comorbid behavioral problems in highly anxious CYP. Previous studies found an association between a stronger attentional proneness to rewarding cues and behavioral problems [52,62]. However, these studies were conducted in young children aged 3–5 and the outcome included both behavioral problems and ADHD symptoms. Therefore, the association between a stronger attentional proneness to rewarding cues and behavioral problems might only be relevant for young children and this may be especially when they also have ADHD. It might be that in adolescence other factors such as encountering stressful life events [63], having low effortful control, lower academic achievement, less parental warmth, low parental monitoring, less good quality of parent-child relationship, delinquent friendships or low housing quality [64] are contributing to behavioral problems or moderate the association between attentional proneness to rewarding cues and behavioral problems. Given our findings, attentional proneness to reward cues does not seem to be a clear risk factor for the development of behavioral problems in highly anxious CYP once they are adolescents.

## Association between behavioral problems in adolescence and anxiety symptoms in young adulthood and moderation/mediation by parental rejection

A second aim of this study was to investigate whether behavioral problems in adolescence increased the risk for having higher anxiety symptoms in young adulthood, whether this association was mediated by parental rejection, and whether this was especially the case for people with high punishment sensitivity.

We did not find evidence for the hypothesis that the association between behavioral problems in adolescence and anxiety in young adulthood would be (partially) mediated or moderated by parental rejection. This is inconsistent with previous findings among high school pupils showing that the prospective relationship between behavioral problems and internalizing symptoms was mediated by both parental rejection and peer rejection [65]. One explanation for this apparent discrepancy in findings might be that parental rejection is especially relevant in early adolescence (as in the study of [65], whereas in young adulthood (as in our study) the influence of parents tends to decrease and peer influence is more crucial [66,67]. We expected that punishment sensitivity would moderate the association between parental rejection and anxiety in young adulthood, however we did not find evidence for this association. This might also be explained by the decreased influence of parents in young adulthood.

In the mediation analysis (where parental rejection was included as mediator) we did find a significant main effect of behavioral problems in adolescence predicting a further increase of anxiety symptoms in young adulthood. However, in the post-hoc moderation analysis (where parental rejection was included as moderator) this main effect of behavioral problems on anxiety symptoms in young adulthood was no longer significant. Therefore, this main effect may be relatively small and future research is needed to assess the robustness of this effect.

## Strengths and limitations

A strength of our study was that our sample was taken from a large prospective cohort study with low sampling bias. Therefore, this sample is well-suited to investigate potential risk factors for psychopathology in CYP. Given the prospective character of the TRAILS study, we were able to not only look at cross-sectional associations but also look at prospective associations. Another strength of the study was that punishment and reward sensitivity were assessed with a behavioral task (SOT) that provides an objective measure that does not require self-understanding or reflection and is therefore also relatively robust against self-representational concerns and demands. Furthermore, the task showed adequate psychometric properties and can differentiate between more automatic processes and processes that allow for some regulatory control. However, a limitation of the use of this task is that it relies on the attentional component of punishment and reward sensitivity. It therefore cannot be ruled out that other components of PS/RS are still relevant risk factors for comorbid behavioral problems in anxious persons. Another limitation was that no diagnostic interviews were conducted to assess clinical diagnoses of the participant in the included assessment waves. We selected the 90th percentile of anxious adolescents and anxious adults, expecting that given prevalence rates of anxiety and the representativeness of our sample, this reflects a clinically anxious group. Preferably, this would have been confirmed using diagnostic interviews. Furthermore, given that we conducted multiple tests, our study is vulnerable for false positives. We wanted to balance the risk of type 1 and type 2 errors and therefore decided to conduct our study with an alpha level of .05. We added Bayesian statistics to provide additional information on the likelihood of the null hypotheses and the alternative hypotheses given our data.

## Conclusion

To conclude, reward and punishment sensitivity as indexed by heightened attentional proneness to general cues of punishment and/or reward do not seem to be risk factors for the development of comorbid behavioral problems in highly anxious CYP. Findings point to comorbid behavioral problems as a potential factor that contributes to the further increase of anxiety symptoms.

## Supporting information

**S1 Fig. Example of screen-setup of the Spatial Orienting Task (SOT).**
(DOCX)

## Author Contributions

**Conceptualization:** Leonie J. Kreuze, Peter J. de Jong, Catharina A. Hartman, Maaike H. Nauta.

**Formal analysis:** Leonie J. Kreuze, Nienke C. Jonker.

**Methodology:** Leonie J. Kreuze, Peter J. de Jong, Nienke C. Jonker, Maaike H. Nauta.

**Project administration:** Leonie J. Kreuze.

**Supervision:** Peter J. de Jong, Maaike H. Nauta.

**Writing – original draft:** Leonie J. Kreuze.

**Writing – review & editing:** Peter J. de Jong, Nienke C. Jonker, Catharina A. Hartman, Maaike H. Nauta.

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
