## [Decision Letter · Decision Letter 0]

21 Jan 2022

PONE-D-21-09707

Title- Behavioral Problems in Anxious Youth: Cross-Sectional and Prospective Associations with Reinforcement Sensitivity and Parental Rejection.

PLOS ONE

Dear Dr. Kreuze,

Thank you for submitting your manuscript to PLOS ONE. After careful consideration, we feel that it has merit but does not fully meet PLOS ONE’s publication criteria as it currently stands. Therefore, we invite you to submit a revised version of the manuscript that addresses the points raised during the review process.

As previously discussed, this is an updated decision letter, revised in order to correct the previous erroneous inclusion of reviewer comments unrelated to the present manuscript. The changes to the letter are limited to these comments, designated as provided by Reviewer 2, and all other aspects of the letter, reviewer comments, and decision are unchanged. This decision letter includes an updated deadline if helpful for your convenience, and thank you for your patience as we addressed this technical issue.   

We look forward to receiving your revised manuscript.

Kind regards,

Vanessa Carels

Staff Editor

PLOS ONE

Journal Requirements:

5. Please upload a copy of Figure S1, to which you refer in your text on page 12. If the figure is no longer to be included as part of the submission please remove all reference to it within the text.

7. We noticed you have some minor occurrence of overlapping text with the following previous publication(s), which needs to be addressed:

- https://pubmed.ncbi.nlm.nih.gov/32445103/

In your revision ensure you cite all your sources (including your own works), and quote or rephrase any duplicated text outside the methods section. Further consideration is dependent on these concerns being addressed.

Reviewers' comments:

Reviewer's Responses to Questions

**Comments to the Author**

1. Is the manuscript technically sound, and do the data support the conclusions?

Reviewer #1: Yes

Reviewer #2: No

2. Has the statistical analysis been performed appropriately and rigorously? 

Reviewer #1: I Don't Know

Reviewer #2: Yes

3. Have the authors made all data underlying the findings in their manuscript fully available?

Reviewer #1: No

Reviewer #2: Yes

4. Is the manuscript presented in an intelligible fashion and written in standard English?

Reviewer #1: Yes

Reviewer #2: Yes

5. Review Comments to the Author

Reviewer #1: The present study examined potential risk factors (e.g., punishment (PS) and reward sensitivity (RS) and parental rejection) for comorbid anxiety and behavioural problems in adolescents. PS/RS were not found to be associated with co-occurring behavioural problems in anxious youth either cross sectionally or prospectively. Behavioural problems predicted anxiety in young adulthood after controlling for adolescent anxiety. Contrary to expected this relationship was not mediated by parental rejection.

This paper has a number of strengths including it longitudinal design and the inclusion of multi method assessment (e.g., self report and behavioural).The study is likely to be of interest to readers of the journal and contributes to our understanding of co-occurring anxiety and behavioural problems which is important given that young people who present with this comorbidity often have a poorer prognosis and response to treatment. Weaknesses of the study include the use of an attentional punishment and reward sensitivity measure only and a lack of a use of diagnostic interviews to assess anxiety and behavioural problems. A few minor comments for discussion below –

I was surprised to see the authors introduce another TRAILS study (Kreuze, Jonker et al., 2020) similarly examining the relationships between punishment sensitivity and reward sensitivity anxiety/behavioural problems in the larger cohort in the discussion on the final page. It is suggested that the authors introduce this study in the introduction and clearly highlight how the present subsequent study is distinct from it and extends upon it to answer a new research question.

To assist with explaining the SOT is it at all possible to include some sort of diagram.

Minor comment

Table 1b reports means for T1, T4 and T5. Was data from Time 1 included in the study or is this a typo?

Reviewer #2: 

The present paper focused on a subsample of children and young people (CYP) with anxiety disorders, examined the prospective association between behavioral problems and anxiety at 6-year follow-up, and the proposed mediation by parental rejection. The study also tested the relationship between reward or punishment sensitivity and behavioral problems. The authors should be commended for the longitudinal design using a database, and considering parenting rejection as an important contextual aspect for adolescent development. However, there are several major concerns that diminish my enthusiasm for the current study.

P4, However, given their high reward expectancy, they are also more prone to detect non-reward (rewards with a lower than expected frequency or lower level of reward), which may lead to anger and oppositional behaviors out of frustrative non-reward.? Besides this logic, there is also another possibility that people with high reward sensitivity display more aggressive or disruptive behavior to get the reward when being hindered.

P5, for the third aim of the current study, I think that parental rejection as a typically harmful environment for adolescents is better to considered as a moderator rather than mediator for the relation between problem behavior in adolescents and later anxiety symptoms in young adulthood. Rather, reward or punishment bias can be served as both moderator or mediator in this relation. I suggest authors run this model in the data analyses.

P22, authors should give more explanation for the significant correlation results.

In short, it is acceptable that the main hypotheses were not found in the current findings. However, authors should have a clear logic about the main variables and give more deep discussion about the current findings.

6. PLOS authors have the option to publish the peer review history of their article (what does this mean?). If published, this will include your full peer review and any attached files.

Reviewer #1: No

Reviewer #2: No

---

## [Author Response · Author response to Decision Letter 0]

3 Mar 2022

The paragraph provide above is the information that is also provided in other papers using the trails data and describes the funding sources. Not all corresponding codes could be found under funding information in the submission programma and were therefore provided in this comment section and added as funders under funding information.

This research is part of the TRacking

Adolescents’Individual Lives Survey (TRAILS). Participating centers

of TRAILS include various departments of the University

Medical Center and University of Groningen, the Erasmus

University Medical Center Rotterdam, the University of Utrecht, the

Radboud Medical Center Nijmegen, and the Parnassia Bavo group,

all in the Netherlands. TRAILS has been financially supported by

various grants from the Netherlands Organization for Scientific

Research (NWO), ZonMW, GB-MaGW, the Dutch Ministry of

Justice, the European Science Foundation, BBMRI-NL, the participating

universities, and Accare Center for Child and Adolescent

Psychiatry. The first author was supported by a PhD fund from the

faculty of Behavioural and Social sciences from the University of

Groningen.

More information on the exact grant number is provided in the paragraph that was provided below this first paragraph.

---

## [Decision Letter · Decision Letter 1]

5 Apr 2022

Title- Behavioral Problems in Anxious Youth: Cross-Sectional and Prospective Associations with Reinforcement Sensitivity and Parental Rejection.

PONE-D-21-09707R1

Dear Dr. Kreuze,

We’re pleased to inform you that your manuscript has been judged scientifically suitable for publication and will be formally accepted for publication once it meets all outstanding technical requirements.

Kind regards,

Sergio A. Useche, Ph.D.

Academic Editor

PLOS ONE

Additional Editor Comments (optional):

Reviewers' comments:

Reviewer's Responses to Questions

**Comments to the Author**

1. If the authors have adequately addressed your comments raised in a previous round of review and you feel that this manuscript is now acceptable for publication, you may indicate that here to bypass the “Comments to the Author” section, enter your conflict of interest statement in the “Confidential to Editor” section, and submit your "Accept" recommendation.

Reviewer #1: All comments have been addressed

Reviewer #2: All comments have been addressed

2. Is the manuscript technically sound, and do the data support the conclusions?

Reviewer #1: Yes

Reviewer #2: Yes

3. Has the statistical analysis been performed appropriately and rigorously? 

Reviewer #1: I Don't Know

Reviewer #2: Yes

4. Have the authors made all data underlying the findings in their manuscript fully available?

Reviewer #1: Yes

Reviewer #2: Yes

5. Is the manuscript presented in an intelligible fashion and written in standard English?

Reviewer #1: Yes

Reviewer #2: Yes

6. Review Comments to the Author

Reviewer #1: This is a revised manuscript of a longitudinal study examining risk factors (e.g., punishment (PS) and reward sensitivity (RS) and parental rejection) for comorbid anxiety and behavioural problems in adolescents. I have reviewed the authors manuscript revisions and they have adequately addressed my comments.

Reviewer #2: I feel satisfied about the authors' revision and have no other comments. Although the main hypotheses were not found in the current findings, the logic and explanation is acceptable.

7. PLOS authors have the option to publish the peer review history of their article (what does this mean?). If published, this will include your full peer review and any attached files.

Reviewer #1: No

Reviewer #2: No

---

## [Editor Report · Acceptance letter]

1 Jun 2022

PONE-D-21-09707R1 

Behavioral Problems in Anxious Youth: Cross-Sectional and Prospective Associations with Reinforcement Sensitivity and Parental Rejection. 

Dear Dr. Kreuze:

I'm pleased to inform you that your manuscript has been deemed suitable for publication in PLOS ONE. Congratulations! Your manuscript is now with our production department. 

Kind regards, 

on behalf of

Dr. Sergio A. Useche 

Academic Editor

PLOS ONE